# Regulation and Roles of Metacyclogenesis and Epimastigogenesis in the Life Cycle of *Trypanosoma cruzi*

**DOI:** 10.3390/pathogens14111149

**Published:** 2025-11-12

**Authors:** Abel Sana, Izadora Volpato Rossi, Marcel Ivan Ramirez

**Affiliations:** 1EVAHPI-Extracellular Vesicles and Host-Parasite Interactions Research Group, Carlos Chagas Institute (Fiocruz-PR), Curitiba 81350-010, PR, Brazil; 2Graduate Program in Cell and Molecular Biology, Federal University of Paraná, Curitiba 81531-970, PR, Brazil; 3Department of Immunology, Parasitology and General Pathology, State University of Londrina, Londrina 86057-970, PR, Brazil

**Keywords:** *Trypanosoma cruzi*, epimastigogenesis, metacyclogenesis, insect vector

## Abstract

*Trypanosoma cruzi*, the etiological agent of Chagas disease, exhibits remarkable developmental plasticity that enables its survival across distinct environments within the insect vector and mammalian host. This review focuses on two critical differentiation processes—metacyclogenesis and epimastigogenesis—emphasising their environmental triggers, metabolic regulation, and roles in parasite transmission and life cycle progression. Metacyclogenesis, occurring in the hindgut of triatomine vectors, transforms replicative epimastigotes into infective metacyclic trypomastigotes and is tightly controlled by factors such as nutrient starvation, pH, and temperature. In contrast, epimastigogenesis allows trypomastigotes to revert to epimastigote forms, primarily in the vector midgut, as part of the parasite’s adaptation to vector colonisation. We compare these processes through the lens of stress-induced signalling and proteomic reprogramming, highlighting their metabolic divergence and ecological significance. Emerging evidence also suggests that extracellular vesicles (EVs) released by different parasite forms may actively modulate these transitions, supporting parasite communication and immune evasion strategies. A better understanding of these transitions provides novel insight into parasite adaptation and reveals potential molecular targets for disrupting the life cycle of *T. cruzi*.

## 1. Introduction

*Trypanosoma cruzi* is an obligate intracellular parasite that alternates between invertebrate and vertebrate hosts, undergoing complex cycles of differentiation from replicative to infective forms [1]. Throughout its life cycle, *T. cruzi* displays multiple morphological stages adapted to specific host environments. In the insect vector, replicative epimastigote forms proliferate free in the midgut, while in mammalian hosts, intracellular amastigotes replicate within cells. Differentiated metacyclic trypomastigotes develop in the posterior region of the insect vector and are responsible for initiating infection upon transmission to mammal hosts. In contrast, bloodstream trypomastigotes—whether naturally circulating or culture-derived—are primarily responsible for systemic dissemination and disease manifestation in mammals, and also contribute to infection of the insect vector [2].

Each of these evolutionary stages possesses a repertoire of surface molecules essential for host–parasite interactions and exhibits distinct metabolic and cellular characteristics that remain incompletely understood [3]. Over the past few decades, a substantial body of research has focused on elucidating the mechanisms underlying *T. cruzi* invasion and persistence during the acute and chronic phases of Chagas disease in mammalian hosts. Parallel efforts have sought to understand how the parasite develops and transitions within the triatomine insect vector [4]. Nonetheless, many gaps persist in our knowledge of the parasite’s life cycle, particularly regarding the molecular and cellular processes that regulate its transition from replicative to infective forms.

Despite the success of vector control campaigns in reducing the incidence of Chagas disease in some endemic countries, several factors have hindered sustained progress. These include increasing migratory flows, lack of disease awareness among healthcare professionals and the public, and insufficient governmental support for long-term control policies. As a consequence, *T. cruzi* transmission persists in endemic regions and has expanded to non-endemic areas, including parts of North America, Europe, and Asia. Moreover, the reemergence of oral transmission over the past three decades has added further complexity to the epidemiological landscape of Chagas disease [5,6].

Importantly, despite the complexity of Chagas disease and the substantial research efforts of past decades, current treatment, mainly benznidazole and nifurtimox, are limited by toxicity, long treatment regimens, and variable efficacy, especially during the chronic phase of infection [7]. No effective vaccine is currently available to prevent *T. cruzi* infection. These limitations highlight the urgent need for novel approaches. In this context, understanding parasite differentiation within the insect vector and the molecular mechanisms governing these processes could offer promising alternative strategies for disease control. Such knowledge may inform new tools to interrupt transmission, especially in endemic regions and in areas where the incidence of Chagas disease is currently rising.

## 2. Biological Differentiation: Epimastigogenesis

In the insect vector (Triatomine), *Tripanosoma cruzi* undergoes two key differentiation events: epimastigogenesis and metacyclogenesis. Epimastigogenesis refers to the process by which bloodstream trypomastigotes, once ingested by the insect and entering its intestinal tract, differentiate into epimastigote forms [8]. It has been proposed that this process begins in the anterior midgut of the insect vector, where blood trypomastigotes initially transform into intermediate forms. These intermediates then migrate to the posterior midgut, where full epimastigote differentiation occurs [1,9]. Indeed, Brener observed a large number of intermediate forms in the anterior midgut of the triatomine 72 and 96 h after infection [10]. Kessler and colleagues found that after 5 days of *R. prolixus* infection, parasites remaining in the anterior midgut still exhibited an amastigote form and no epimastigotes were found in this portion of the intestine [11]. Recent findings suggest that the anterior midgut may be inhospitable to bloodstream trypomastigotes, as parasite numbers decrease drastically within 24 h of infection [9]. Perhaps as a survival mechanism, surviving parasites differentiate and rapidly migrate to the posterior midgut where they complete epimastigogenesis. Although some studies propose that there is a direct differentiation from trypomastigotes to epimastigotes, in vitro data support the existence of intermediate forms before reaching epimastigotes. Silva and collaborators, working with the EPm6 strain of *T. cruzi*, observed that between 3 and 6 days of differentiation, parasites exhibit rounded morphology without flagella before transforming into epimastigotes [8]. Graterol and colleagues observed that these rounded forms are biologically and antigenically equivalent to the amastigote stage [12]. This rounded morphology was also observed by Kessler et al. in the Dm28c strain at 24 h post-differentiation; only after 48 h did parasites exhibit the elongated flagellum characteristic of epimastigotes [11].

Despite this evidence, there remains no consensus on the precise sequence of events. A summary of models and technical challenges for analyzing epimastigogenesis is provided in Box 1.

Blood trypomastigotes exist in two morphological types: (a) slender forms, with a long and narrow nucleus, a very short free flagellum, a discrete undulating membrane and a kinetoplast far from the posterior end; (b) broad forms: possessing round nuclei, a kinetoplast close to the posterior end, long flagella, and well-developed undulating membranes e [13]. In contrast, epimastigotes have a disk-shaped kinetoplast located near the nucleus, abundant cytoplasm, a short undulating membrane and a well-developed flagellum. In trypomastigotes, the kinetoplast is positioned posteriorly, whereas in epimastigotes, it is located anteriorly in relation to the nucleus [13,14]. Therefore, the transformation of blood trypomastigotes to epimastigotes involves substantial remodelling in the shape of the cell body, positioning of the kinetoplast and flagellum in relation to the nucleus.

A summary of the main differences between blood trypomastigotes and epimastigotes is listed in Table 1, and the main experimental models used to study epimastigogenesis is listed in Table 2.

Box 1Technical challenges in analyzing differentiation in *T. cruzi. 1.*
**In Vitro vs. In Vivo *Models.***In vitro and in vivo methods have been used to study epimastigogenesis (Table 2). The study of in vitro differentiation of trypomastigotes to epimastigotes is based on the use of protocols with different incubation conditions of the parasite and analysis of morphological, biochemical and functional changes during differentiation. In vivo methods generally involve feeding triatomine insects with trypomastigote forms and analysis of parasite development in the animal’s digestive tract. The first changes seen in the parasites are the repositioning of the kinetoplast closer to the nucleus and a decrease in the size of the flagellum, until the flagellum is completely disassembled and the parasite exhibits an intermediate amastigote-like form. Subsequently, the flagellum grows back, producing a typical elongated epimastigote flagellum, and then cell division begins [8,12,15,16]. Different studies show that this process can take approximately 2 to 12 days, depending on the differentiation protocol. The extensive death of blood trypomastigotes in the anterior midgut of the insect vector at the beginning of the infection [9] is a challenge in the study of epimastigogenesis in vivo, since it can make it difficult to analyze the differentiation rate. The study of changes in gene expression, protein and lipid profile, resistance to complement-mediated lysis and the ability to infect mammalian cells becomes difficult due to the small number of epimastigotes, as a consequence of the death of trypomastigotes. Another challenge is the large number of insects required to obtain representative sample quantities. In addition, counting *T. cruzi* forms at different times after differentiation involves killing the insects and dissecting parts of the intestine [9]. In vitro studies, although they help to solve the problem of the number of epimastigotes, also present their challenges. Perhaps the biggest challenge is the impossibility of simulating what actually happens in epimastigogenesis in the natural environment, taking into account the role of factors inherent to the insect such as nutritional status, defense systems and microbiota. Another challenge is the need for large numbers of cell culture trypomastigotes (TCTs) to obtain the desired number of epimastigotes. Furthermore, not all parasites transform into epimastigotes at the same time, so replication may begin before the population is fully differentiated.

**Table 1 pathogens-14-01149-t001:** Morphological and Biological Differences.

Feature	TCT (Trypomastigote)	Intermediate Form	Epimastigote
**Cell Shape**	Slender, fusiform	Pleomorphic, transitioning	Elongated, more robust
**Flagellum Position**	Free flagellum, posterior kinetoplast	Partial retraction of flagellum	Emerges near nucleus, anterior kinetoplast
**Kinetoplast Position**	Posterior to nucleus	Intermediate	Anterior to nucleus
**Motility**	Highly motile	Variable motility	Actively motile (in culture)
**Replication**	Non-replicative in vertebrate host	Limited or no replication	Replicative (in vector or culture)
**Infectivity**	Highly infective to mammalian cells	Variable infectivity (some partially infective)	Traditionally non-infective (except rdEpi)
**Surface Markers**	High expression of trans-sialidase, mucins	Mixed expression	Express cruzipain, GP72
**Complement Sensitivity**	Complement-resistant	Partially resistant	Sensitive (except rdEpi)
**Occurrence**	Bloodstream or tissue of mammalian host	During transformation (e.g., in vitro epimastigogenesis)	Vector midgut, axenic culture
**Role in Life Cycle**	Infection initiation	Transitional, adaptive stage	Replication in vector; precursors of metacyclics

**Table 2 pathogens-14-01149-t002:** Methods of study and characterization of epimastigogenesis.

Method	Strain/Clone	From Which Evolutionary Form? (Blood Tripe, Culture Tripe, Metacyclic)	Morphological Characterization? (e.g., Giemsa, Immunofluorescence)	Biochemical Characterization? (Protein Expression, Proteomics, Markers, PCR)	Functional Characterization? (Replication, Invasion, Complement System Assays, etc.)	Reference
In vitro: Incubation in LIT medium with varying oxygen tension (high and low) at different times (1, 2, 4, 6, 8, 11, 12 days)	Clone EPm6 (isolated from human)	Tissue-culture derived trypomastigotes	Microscopy (Giemsa)	Metabolic analysis (glucose, ammonia and pH of the medium)	Replication (growth curve)	[15]
In vitro: Semi-defined medium containing L-proline (MEMTAUHLA) (0 to 16 days) and low oxygen content	EPm6 Clone (isolated from human)	Tissue-culture derived trypomastigotes	Microscopy (Giemsa)	Peptidic and antigenic profiling (immunoblotting)	Replication (growth curve)	[8]
In vitro: ML15-HA medium (insect embryonic cell culture medium) (0 to 10 days)	Clone Dm28c	Tissue-culture derived trypomastigotes	Microscopy (Giemsa)	Peptidic and antigenic profiling (immunoblotting)	Replication (growth curve)	[12]
In vitro: LITB medium (0 to 8 days) with different oxygen levels	Clone EPm6 (isolated from human)	Tissue-culture derived trypomastigotes	Microscopy (Giemsa)	Antigenic profile (stage-specific markers of amastigotes)	Replication (growth curve); Complement system resistance	[12]
In vivo: Feeding of *Rhodnius prolixus* with cultured trypomastigotes (up to 10 days after infection)	Clone EPm6, Strain DmN5 and Strain RpN2	Tissue-culture derived trypomastigotes	Microscopy (Giemsa staining)	N/A	N/A	[16]
In vivo: feeding of *Rhodnius prolixus* with blood trypomastigotes (up to 15 days after infection)	CL Strain	Bloodstream trypomastigotes	Microscopy (Giemsa staining)	N/A	Migration and differentiation	[17]
In vitro: LITB medium (0 a 5 days)	Dm28c clone	Tissue-culture derived trypomastigotes, metacyclic	Immunofluorescence	Protein expression, proteomics	Invasion assays, complement system resistance	[11]
In vivo: feeding of *Rhodnius prolixus* with blood trypomastigotes; anterior midgut (AM) and posterior midgut (PM) were individually dissected and used for immunofluorescence analysis of *T. cruzi* forms.	Clone Dm28c	Blood trypomastigotes	Immunofluorescence	Protein expression	Invasion assays, complement system resistance	[11]

N/A means Not Applicable.

## 3. Biological Differentiation: Metacyclogenesis

Metacyclogenesis: From Epimastigotes to Infective Forms. The second critical differentiation process in the insect vector is metacyclogenesis, where epimastigotes transform into metacyclic trypomastigotes: the infective forms responsible for initiating mammalian infections [18]. This differentiation stage of *T. cruzi* has generated interest because it involves the transformation of a non-infective form (epimastigote) into an infective form (metacyclic trypomastigote). This process occurs in the triatomine hindgut (rectum), where after epimastigotes migrate from the midgut, they adhere through their flagella to the epithelium and differentiate [14,17].

Metacyclogenesis is marked by several phenotypic and cellular transformations, including: Chromatin remodelling, Repositioning of the kinetoplast to a posterior location, Elongation of the cell body, Reduction of the reservosome, Transition to a spherical kinetoplast morphology, Narrowing of the nucleus and overall cell shape [14,17].

### In Vitro Models of Metacyclogenesis

Although metacyclogenesis occurs naturally in the vector, it can be replicated in vitro under axenic conditions using media that simulate the chemical properties of insect urine [19]. These conditions, particularly the TAU medium and its derivatives, have enabled the reproducible and high-yield induction of metacyclogenesis in the laboratory. For example: (i) Contreras et al. achieved ~85% conversion to metacyclic trypomastigotes within six days using TAU medium supplemented with 10% newborn calf serum [19]. (ii) Gonçalves et al. used the enriched TAU3AAG medium (with added glutamate, proline, aspartate, and glucose) and observed 39.16% metacyclic forms after 48 h, 45.6% after 72 h [14]. These studies have greatly expanded our understanding of the molecular and metabolic regulation of differentiation and are considered essential tools for studying infectivity and stage conversion.

## 4. Environmental Cues: Temperature, Stress, and Nutrient Sensing

Several factors act as critical triggers for *T. cruzi* differentiation. Factors such as nutritional stress, parasite strain, insect vector species, host mammal serum, metabolic stress, pH, temperature, vector hemolymph components, osmolarity and cyclic AMP (cAMP) are identified as important inducers of metacyclogenesis [2]. The permissiveness of the insect vector’s intestine to the development of *T. cruzi* largely depends on the nutritional status of the host, the genetic and phenotypic characteristics of the parasite strain, the presence of trypanolytic factors, digestive enzymes, lectins, resident microbiota, and the insect’s endocrine regulation. The rate of metacyclogenesis is affected by the nutritional status of the vector, since the lack of nutrients in the intestinal tract of the vector affects the population of *T. cruzi*, mainly epimastigotes and metacyclic forms [2]. High yields of metacyclic forms can be achieved by initially placing epimastigotes in a nutrient-depleted stress medium devoid of sugars, amino acids, and proteins, commonly referred to as artificial triatomine urine (TAU), followed by transfer to the same medium enriched with L-proline, glutamic acid, aspartic acid, and glucose. Homsy et al., working with a Peruvian isolate clone, demonstrated that L-proline and L-glutamate significantly enhanced metacyclogenesis after four days, whereas leucine and isoleucine (10 mM) inhibited differentiation [20].

Gonçalves et al. subjected epimastigotes to a 2 h nutritional stress period in TAU medium, then induced differentiation in TAU3AAG (TAU supplemented with specific amino acids and glucose). The parasites subjected to stress exhibited wrinkled surfaces and twisted cell bodies, and higher numbers of metacyclic trypomastigotes were obtained after 72 h of incubation [14]. Another important factor for metacyclogenesis is temperature. Lower temperature, for example, 20 °C instead of 28 °C, can delay metacyclogenesis [2].

The finding that cyclic AMP levels are higher in metacyclic trypomastigotes than in epimastigotes suggests that metacyclogenesis is stimulated by cAMP or its analogues and adenylate cyclase activators [21]. Gonzales-Perdomo studied the effects of cAMP and its analogues (Dibutyryl cyclic AMP and 8-Bromoadenosine 3′,5′-cyclic monophosphate) on metacyclogenesis in TAU medium supplemented with glucose (a condition in which differentiation does not occur). The results showed a significant increase in the amount of metacyclic forms, mainly in the treatment with 8-Bromoadenosine 3′,5′-cyclic monophosphate [22]. Fraidenraich et al. further confirmed the role of cAMP by isolating a ~10 kDa peptide from the hindgut of *Triatoma infestans* that activates adenylate cyclase and promotes metacyclogenesis in vitro [23]. In contrast to metacyclogenesis, the cues that initiate epimastigogenesis remain poorly characterised. One of the few known factors is the sudden temperature drop experienced by blood trypomastigotes upon ingestion by the insect vector. This decrease in temperature is believed to initiate transformation, both in vivo and in vitro [8]. Sana et al. compared the transformation of cell culture-derived trypomastigotes (TCTs) at 28 °C versus 37 °C in *T. cruzi* strains CL Brener and Dm28c and found that epimastigote formation occurred exclusively at 28 °C, with visible transformation observed after 72 h [24]. Other contributing factors include oxygen tension, parasite density, and components of the insect digestive tract, such as digestive enzymes, antimicrobial peptides, and the gut microbiota [25]. For example, low oxygen levels and high parasite density have been reported to accelerate differentiation [12]. However, the exact signalling pathways triggered by these environmental factors, the gene networks involved, and the interplay between parasite and microbiota remain largely unknown, representing important avenues for future research.

## 5. Intestinal Colonization by *T. cruzi*: Vector Resistance and Parasite Adaptation

During its passage through the digestive tract of triatomine insects, *T. cruzi* encounters a variety of environmental stressors that act as signals for survival and differentiation. These stressors include pH fluctuations, temperature changes, nutrient depletion, osmotic imbalances, and oxidative stress. The parasite’s response to these cues involves not only morphological changes but also extensive metabolic and molecular reprogramming, enabling it to survive hostile conditions and enhance its infective potential.

Upon ingestion with the blood meal, *T. cruzi* is exposed to multiple components within the anterior and posterior midgut of the insect vector. These include blood digestion products such as hemolytic factors, α-D-globin-derived peptides and lectins, all of which can modulate the dynamics of colonization, multiplication and transformation of *T. cruzi* [26,27,28,29]. In this context, the parasite must also confront oxidative stress, primarily caused by reactive oxygen species (ROS) generated during haemoglobin degradation, which releases large quantities of heme into the midgut [30,31]. To counteract these toxic conditions, *T. cruzi* utilises a unique antioxidant system centred on trypanothione and its corresponding enzyme, trypanothione reductase, in contrast to other eukaryotes, which typically rely on glutathione or thioredoxin-based systems. The redox state of the vector midgut plays a decisive role in determining the parasite’s survival, replication, and differentiation. For instance, heme supplementation has been shown to promote epimastigote proliferation while impairing metacyclogenesis, whereas antioxidant treatment (e.g., N-acetylcysteine) enhances the formation of metacyclic trypomastigotes, both in vitro and in vivo [32]. The presence of high levels of superoxide molecules in the hemolymph of *R. prolixus* correlates with an increase in the death and elimination of *T. cruzi* (when injected into the hemolymph) [33]. This observation helps explain why *T. cruzi* remains confined to the intestinal tract and does not colonise the haemolymph, unlike *Trypanosoma rangeli*, which is capable of systemic invasion.

### 5.1. Immune Responses of the Vector and T. cruzi Persistence

One of the key determinants for successful vector infection is the insect’s innate immune response, which involves a variety of defence mechanisms. These include phagocytosis, reactive oxygen and nitrogen intermediates, components of the phenoloxidase cascade, as well as the action of lysozymes, defensins, lectins, and other antimicrobial effectors [33,34,35,36,37,38,39,40].

Collectively, these responses are capable of eliminating up to 80% of invading *T. cruzi* parasites [9]. However, despite their effectiveness, these immune defences are insufficient to fully prevent the establishment of *T. cruzi* in the insect midgut, allowing the parasite to successfully colonise and persist within the vector.

The ability of *T. cruzi* to persist in its insect hosts and its gain in infectivity depend on the tight regulation of protein expression (discussed in the next section). Understanding the molecular underpinnings of these transitions, such as signaling pathways, metabolic changes, and expression of virulence factors, may reveal new targets for intervention and/or interruption of the *T. cruzi* life cycle, contributing to the control of transmission.

### 5.2. Biochemical and Molecular Reprogramming

Following ingestion by the insect vector, *T. cruzi* bloodstream trypomastigotes differentiate into epimastigotes. These forms are metabolically active and proliferative, primarily relying on glucose but also utilising amino acids and lipids as energy sources [41,42,43,44,45].

Epimastigotes exhibit metabolic plasticity, switching to amino acid catabolism via oxidative phosphorylation upon glucose depletion [46,47,48,49].

Transcriptomic analyses reveal upregulation of genes related to the Krebs cycle, respiratory chain, oxidative phosphorylation, and macromolecule synthesis [50]. Genes from the pentose phosphate pathway are also elevated, supporting NADPH production and nucleotide biosynthesis.

Additionally, fermentation-related enzymes such as acetaldehyde and alcohol dehydrogenases are upregulated, reflecting the parasite’s adaptability to varying oxygen and nutrient conditions [50].

In the midgut of triatomines, epimastigotes divide repeatedly by binary fission and may adhere to the perimicrovillar membranes of intestinal cells. In the rectum, a proportion of epimastigotes adhere to the rectal cuticle and transform into metacyclic trypomastigotes, which are eliminated in feces and urine [2,51,52]. Little is known about the parasite surface components that are important for this adhesion, but it is believed that this binding is due to the abundant glycoinositol phospholipids (GIPLs) molecules in the epimastigote plasma membrane [53,54,55]. For example, deletion of the gene encoding a surface glycoprotein (GP72) of epimastigotes led to a significant decrease in the parasite population in the vector [56,57].

### 5.3. Metabolic Shifts and Autophagy as Pre-Adaptive Events in Metacyclogenesis

Epimastigotes in exponential growth differ significantly from those in the stationary phase. Morphologically, stationary forms are more elongated [58] and undergo a metabolic switch from glycolysis to amino acid catabolism [59,60,61]. Transcriptional activity drops in the stationary phase, although mRNA half-lives increase, likely reflecting energy-saving adaptations for stress survival [62].

In this stationary phase, although the abundance of mRNAs is reduced, their half-lives increase three to seven times [63], suggesting a possible energy-saving mechanism that favors survival under conditions of prolonged stress. These and other observations have pointed to the stationary phase of epimastigotes as a pre-adaptive stage for metacyclogenesis [64,65]. Although the stationary phase was defined based on an observation of the growth curve of epimastigotes in vitro, it has biological relevance in natural environments. In the vector’s intestine, where nutrients are scarce between blood meals, epimastigotes face a condition of nutritional scarcity, a triggering factor for metacyclogenesis.

### 5.4. Autophagy and the Regulation of Metacyclogenesis

Several environmental cues such as nutrient deprivation, altered pH and osmolarity, oxygen tension, and adhesion to the intestinal epithelium, trigger *T. cruzi* metacyclogenesis. Although the full molecular mechanisms remain unclear, autophagy has emerged as a central regulatory process during this differentiation [66,67,68,69].

Under nutrient-limited conditions, levels of polyamines like spermidine and spermine increase, leading to the inhibition of histone acetyltransferases (HATs). This inhibition promotes transcription of Atg genes (autophagy-related proteins) and induces autophagosome formation [66,67].

At the molecular level, *T. cruzi* encodes homologues of the eukaryotic autophagy machinery, including TcVps34, a class III phosphatidylinositol-3-kinase, and its regulatory partner TcVps15. These form a functional complex responsible for generating phosphatidylinositol-3-phosphate (PI3P) at the initiation sites of autophagosomes [67,70]. The TcVps34–TcVps15 complex is essential for the recruitment of TcAtg8.1 and TcAtg8.2, which mediate membrane elongation and cargo sequestration [67,71,72]. Pharmacological inhibition of this complex using wortmannin or 3-methyladenine significantly decreases autophagosome formation and arrests parasite differentiation, underscoring the dependence of metacyclogenesis on autophagic flux [66,72].

Metacyclogenesis and epimastigogenesis are adaptive differentiation processes in *T. cruzi* governed by conserved mechanisms. Both are induced by environmental stressors such as nutrient deprivation and pH shifts, relying on autophagy and metabolic reprogramming for organelle and surface remodelling. Metacyclogenesis in the insect vector yields infective metacyclic trypomastigotes, whereas epimastigogenesis produces replicative epimastigote forms. Together, they reflect *T. cruzi*’s developmental plasticity and its reliance on autophagy-linked pathways to adapt across host and vector environments.

### 5.5. Molecular and Structural Reprogramming During Metacyclogenesis

During *T. cruzi* metacyclogenesis, stage-specific genes linked to survival and mammalian infectivity are expressed before morphological changes occur [73,74]. Some genes are transiently expressed [75,76] and large-scale phosphoproteomic analyses have identified over 260 modulated phosphorylation sites during differentiation [76]. These changes include not only the protein profile [77,78] but also membrane lipid composition [79,80] and surface glycoconjugates [81].

The most notable changes occur at the parasite’s surface. Proteins such as trans-sialidases, MASPs, gp82, calpain, cruzipain, and DGF-1 are differentially expressed and linked to metacyclic infectivity [76,82]. Increased expression of flagellar adhesion glycoproteins (FAGs) reinforces the role of adhesion in this process. Cytoskeletal proteins, including subpellicular microtubules, motor proteins, and flagellar components, are also reprogrammed, aligning with the parasite’s morphological changes [83].

Since *T. cruzi* regulates gene expression mainly post-transcriptionally, protein expression is more closely associated with translational control than with mRNA abundance [84]. Translation of maintenance proteins is downregulated, while virulence factor synthesis is enhanced. Proteins involved in gene transcription and translation are differentially phosphorylated, although global transcription remains relatively stable until late metacyclogenesis [85]. RNA-binding proteins (RBPs) play a key role in regulating mRNA fate and translation efficiency [86,87], while mRNA granules protect transcripts and support the parasite’s adaptability under stress [88,89].

## 6. Parasite–Vector Interaction and Transmission Dynamics

The transmission of *T. cruzi* is intricately linked to its development within triatomine insect vectors, particularly *Rhodnius prolixus*, *Triatoma infestans*, and *Panstrongylus megistus*. The establishment of *T. cruzi* in triatomines is determined by the parasite strain and triatomine species. Within a single vector (with mixed infections), *T. cruzi* strains can behave differently. In mixed infections of a *T. infestans* population, some strains do not develop in all vectors, while in others only a small number develop, but they have a lower percentage of metacyclic trypomastigotes [90,91]. Paranaiba et al. evaluated the susceptibility of Triatoma infestans to different strains of *T. cruzi* and observed that after 49–50 days post-infection, the percentage of nymphs with parasites in the midgut ranged from 15 to 25%: 15% for strain CL Brener, 20% for DM28c, and 25% for strains YUYU and Bug2145cl10 [92]. These findings demonstrate that *T. cruzi* strains can present different profiles within the same triatomine species.

Successful differentiation of the parasite into its infective form, the metacyclic trypomastigote, hinges on a range of physiological and environmental variables within the insect’s digestive tract. Among these, the alternation between feeding and starvation cycles, fluctuations in gut pH, the triatomine immune response, and ambient temperature exert significant regulatory effects on parasite development within the vector.

### 6.1. Feeding and Starvation Cycles

Triatomines are obligate haematophagous insects capable of enduring extended fasting periods, which directly impact the internal ecology of the gut and the fate of resident parasites. Studies have demonstrated that in *R. prolixus*, *T. infestans*, and *Panstrongylus megistus*, prolonged starvation leads to a decline in *T. cruzi* populations, with reduced survival particularly notable in the hindgut where metacyclic forms typically accumulate [93,94,95,96]. These conditions alter the gut microbiota and mechanical flow, impairing the environment necessary for parasite differentiation. In parallel, starvation induces metabolic stress in the vector, increasing oxidative pressure and potentially compromising the transformation of epimastigotes into infective forms [97].

### 6.2. pH Gradients and Immune Responses in the Vector

The triatomine digestive tract is characterized by regional variations in pH, which are critical for enzymatic activity and influence the adhesion and development of *T. cruzi*. *T. infestans* and *Panstrongylus* species exhibit pH environments that can support parasite adaptation, partially through modulation of surface proteins implicated in host cell recognition [98,99]. Concurrently, the vector’s immune system deploys innate defense mechanisms, such as phenoloxidase activation and antimicrobial peptide synthesis, which fluctuate depending on the nutritional and developmental status of the insect [39,100]. These physiological differences may explain variations in vector competence across triatomine species.

### 6.3. Temperature and Feeding Frequency

Temperature exerts a dual influence by shaping both vector physiology and parasite development. In *T. infestans* and *Panstrongylus geniculatus*, temperature shifts affect blood digestion kinetics, immune responsiveness, and vector survival. Warmer conditions tend to accelerate development but at the cost of higher mortality and reduced energy reserves [101]. Feeding frequency is similarly critical: repeated blood meals provide consistent nutrients that support parasite replication, while longer intervals between feedings can lead to decreased metacyclic densities in the rectum [102].

### 6.4. Parasite-Induced Modulation of Vector Physiology

There is growing evidence that *T. cruzi* does not merely inhabit the vector gut passively but actively alters host physiology to its advantage. Infections in *R. prolixus*, *T. infestans*, and *Panstrongylus* spp. have been associated with changes in mitochondrial activity and epithelial homeostasis, favoring parasite persistence and transmission [103]. Moreover, extracellular vesicles released by the parasite have been shown to modulate midgut cellular responses and may play a role in shaping the gut environment [104]. Notably, these interactions appear to be strain-specific, suggesting that certain parasite lineages may be more adept at exploiting vector physiology for transmission success. Sana and collaborators observed that extracellular vesicles released by *T. cruzi* strain CL Brener after 24 and 72 h of in vitro epimastigogenesis induced the formation of epimastigotes, suggesting that extracellular vesicles may be modulating both the parasite and the vector, favoring differentiation [24].

Understanding the nuanced and multifaceted nature of these interactions, particularly those involving *T. infestans* and *Panstrongylus megistus*, which are key vectors in domestic and sylvatic cycles across Latin America, is essential for advancing vector control strategies and improving models of disease transmission. The convergence of physiological, environmental, and molecular factors in shaping vector competence offers both challenges and opportunities for Chagas disease management.

## 7. Genetic Diversity: Relationship with the Development of *T. cruzi* in the Vector Insect

In addition to the aspects discussed above, it is important to emphasise that *T. cruzi* is an exceptionally genetically diverse protozoan. Based on molecular analyses, strains are classified into seven discrete typing units (DTUs), designated TcI–TcVI and TcBat [90,105]. This genetic diversity has been linked to differences in parasite differentiation, survival, and infection establishment within the insect vector. Studies using *Rhodnius prolixus* and *Triatoma infestans* have reported developmental and infective variations among distinct DTUs. For example, TcI strains can complete their development in *R. prolixus*, whereas TcII strains are eliminated; however, both genotypes successfully develop in *T. infestans* [38,105,106]. When evaluating *T. cruzi* distribution along the intestinal tract of *T. infestans*, Paranaiba et al. observed higher parasite densities in the rectum and increased numbers of metacyclic trypomastigotes excreted in the vector’s urine for the Bug (TcV) and YuYu (TcI) strains, compared with CL-Brener and Dm28c [92]. The initial establishment of infection also depends on a complex interplay between parasite and vector determinants. For instance, 24 h after the infection of R. prolixus with the CL strain (DTU TcVI), a marked reduction in circulating trypomastigotes is observed, and after four days, parasites are no longer detected in the stomach [90].

The interaction of the parasite with insect tissues varies according to genotype and developmental stage. The biochemical and morphological transformations that *T. cruzi* undergoes within the vector’s intestinal tract occur through dynamic and stage-specific interactions with insect structures. During metacyclogenesis, epimastigotes located in the rectum attach to the rectal cuticle, initiating their differentiation into metacyclic trypomastigotes [105]. This attachment is mediated by epimastigote surface molecules, including glycoinositol phospholipids (GiPLs) and genotype-specific glycoproteins, which differ among *T. cruzi* lineages [105,107].

That evidence highlights that the compatibility between *T. cruzi* lineages and triatomine species is not uniform but rather shaped by a highly complex relationship. Such interactions may involve factors such as surface molecule recognition, immune responses in the insect gut, and microbiota composition, all of which can influence parasite survival and differentiation. Therefore, understanding the genetic diversity of *T. cruzi* and its influence on parasite–vector interactions is essential for elucidating the dynamics of this relationship across different ecological contexts. Comprehensive studies integrating genomic, transcriptomic, and vector biology approaches are crucial to uncover the molecular mechanisms underlying these genotype-specific compatibilities and to support the development of effective Chagas disease control strategies.

## 8. Concluding Remarks

Understanding *T. cruzi* epimastigogenesis and metacyclogenesis is fundamental to elucidating the biology of the parasite’s developmental stages within the insect vector. Epimastigogenesis leads to the transformation of *T. cruzi* into its replicative form (epimastigote), allowing the parasite to be maintained in its invertebrate host, whereas metacyclogenesis gives rise to the infective metacyclic trypomastigote that resumes the mammalian phase of the life cycle. Despite notable progress in characterising these differentiation processes, particularly metacyclogenesis, many aspects remain unresolved. Which surface molecules of trypomastigotes and epimastigotes determine the efficiency of epimastigogenesis and metacyclogenesis, respectively? How do intestinal factors of triatomine vectors—such as microbiota, lectins, proteases, and immune components—modulate parasite survival and differentiation at the molecular level? What mechanisms enable each stage of *T. cruzi* to evade harmful compounds within the triatomine gut? How do alterations in the insect microbiota mechanistically influence both differentiation processes? Do the molecular dynamics regulating epimastigogenesis vary among triatomine species? Furthermore, how are parasite–parasite and parasite–vector communication mediated by extracellular vesicles, shape survival, migration, and differentiation? Addressing these questions will deepen our understanding of *T. cruzi*’s interaction dynamics within its invertebrate host and may ultimately inform the development of strategies to disrupt parasite transmission in the vector.

## Data Availability

The original contributions presented in this study are included in the article. Further inquiries can be directed to the corresponding author.

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
