# Peer review of "Regulation and Roles of Metacyclogenesis and Epimastigogenesis in the Life Cycle of Trypanosoma cruzi"

_pathogens, 2025, doi:10.3390/pathogens14111149_

Round 1
Reviewer 1 Report
Comments and Suggestions for Authors
In this manuscript, the authors present a review of Trypanosoma cruzi differentiation during its life cycle, focusing on the changes that occur in insect vectors: epimastigogenesis and metacyclogenesis. They conducted an extensive literature review and primarily describe the morphological and physiological changes that occur during these transitions, as well as the environmental and molecular conditions and events that could be involved in these two differentiation processes. This manuscript represents a valuable contribution to the field.
I have several concerns and suggestions that I believe could improve the follow-up and understanding of the article by interested readers.
Major considerations:
- Include a final/concluding remarks section at the end of the manuscript.
- Since the manuscript analyzes results obtained from different strains of T. cruzi, it would be relevant to include a comment on how the high genomic diversity of this parasite may determine its ability to differentiate within the insect vector, whether from a specific strain or isolate.
- There are some issues with the organization of the manuscript that should be addressed: 1) “The Box: Technical challenges in analyzing differentiation in T. cruzi” and the following subtitle “In vitro vs. In vivo models” could simply be another section of the manuscript; 2) Some headings (or subheadings) could be more specific based on what is discussed in those sections. For example, in the section “Biological Differentiation and Morphogenesis” (line 71) was discussed only about epimastigogenesis. The same goes for the subheading on line 233. Perhaps the authors could try organizing the article into general aspects of insect and parasitic physiology, and then into specific aspects of epimastigogenesis and metacyclogenesis.
Minor considerations:
- The article by Sana et al. (2025) should be cited, where the role of extracellular vesicles (EVs) in modulating vector physiology was discussed (lines 361-364). Especially since the abstract even included the importance of parasite EVs for T. cruzi differentiation in the insect vector.
- Please review the text to see if T. cruzi is italicized.
- I suggest expanding the comment (lines 344-346) on the importance of the relationship between intrinsic differences in different triatomine species and the parasite's ability to differentiate within the insect with respect to vector competence.
- If possible, expand the discussion on the relationship between the insect microbiota and T. cruzi differentiation.
- A period is missing after "thioredoxin-based systems" (line 206).
- The reference to Table 1 was not included in the text.
- There are several capitalized words in lines 121–123 that should begin with lowercase letters.
Author Response
Dear Editor and Reviewers,
We would like to sincerely thank you for the time and effort dedicated to reviewing our manuscript entitled “Extracellular Vesicles Promote Epimastigogenesis in Trypanosoma cruzi.” We greatly appreciate the constructive comments and suggestions provided, which have been invaluable in improving the clarity, structure, and scientific quality of our work.
Following the reviewers’ recommendations, we have thoroughly revised the manuscript. Major changes include a clearer description of the experimental design and statistical analyses; expanded discussion linking extracellular vesicle signalling to differentiation mechanisms; Improved English language and figure labelling throughout the text.
Below, we provide a detailed, point-by-point response to each comment, indicating how and where the manuscript has been modified. All changes have been highlighted in the revised version for ease of reference.
We hope that the revised manuscript now meets the journal’s standards and addresses all reviewers’ concerns. We remain at your disposal for any further clarification.
With our best regards,
Marcel I Ramirez, PhD on behalf of all authors
Instituto Carlos chagas
Curitiba
Brazil
Reviewer 1
Comments and Suggestions for Authors
In this manuscript, the authors present a review of Trypanosoma cruzi differentiation during its life cycle, focusing on the changes that occur in insect vectors: epimastigogenesis and metacyclogenesis. They conducted an extensive literature review and primarily describe the morphological and physiological changes that occur during these transitions, as well as the environmental and molecular conditions and events that could be involved in these two differentiation processes. This manuscript represents a valuable contribution to the field.
I have several concerns and suggestions that I believe could improve the follow-up and understanding of the article by interested readers.
Major considerations:
Include a final/concluding remarks section at the end of the manuscript.
Since the manuscript analyzes results obtained from different strains of T. cruzi, it would be relevant to include a comment on how the high genomic diversity of this parasite may determine its ability to differentiate within the insect vector, whether from a specific strain or isolate.
Answer: We thank the reviewer for this valuable suggestion. We have included a final section entitled “Concluding Remarks”, which integrates our key findings and perspectives. Additionally, a new subsection has been added to discuss the genetic diversity of T. cruzi and how it influences the parasite’s developmental capacity within the insect vector.
There are some issues with the organization of the manuscript that should be addressed:
1) “The Box: Technical challenges in analyzing differentiation in T. cruzi” and the following subtitle “In vitro vs. In vivo models” could simply be another section of the manuscript;
We appreciate the suggestion. We chose to maintain this content as a boxed section rather than a main section, as it provides complementary information that, while not essential to the core discussion, offers useful technical insights for readers.
2) Some headings (or subheadings) could be more specific based on what is discussed in those sections. For example, in the section “Biological Differentiation and Morphogenesis” (line 71) was discussed only about epimastigogenesis. The same goes for the subheading on line 233. Perhaps the authors could try organizing the article into general aspects of insect and parasitic physiology, and then into specific aspects of epimastigogenesis and metacyclogenesis.
Thank you for the observation. We have revised several subheadings to make them more specific and reflective of their content. The structure now moves from general physiological aspects to specific differentiation processes.
Minor considerations:
The article by Sana et al. (2025) should be cited, where the role of extracellular vesicles (EVs) in modulating vector physiology was discussed (lines 361-364). Especially since the abstract even included the importance of parasite EVs for T. cruzi differentiation in the insect vector.
I suggest expanding the comment (lines 344-346) on the importance of the relationship between intrinsic differences in different triatomine species and the parasite's ability to differentiate within the insect with respect to vector competence.
If possible, expand the discussion on the relationship between the insect microbiota and T. cruzi differentiation.
We thank the reviewer for these careful observations. All corrections have been implemented. The article by Sana et al. (2025) has been cited as recommended. The “Parasite–Vector Interaction and Transmission Dynamics” section now includes expanded discussion on vector species-specific differences and microbiota influences. Typographical issues, table references, and capitalisation have been corrected.
Please review the text to see if T. cruzi is italicized.
A period is missing after "thioredoxin-based systems" (line 206).
The reference to Table 1 was not included in the text.
There are several capitalized words in lines 121–123 that should begin with lowercase letters.
The text has been revised and the indicated corrections have been made. A reference to table 1 has been included at the end of the section “Biological Differentiation: Epimastigogenesis”.
Reviewer 2 Report
Comments and Suggestions for Authors
The aim of this study has been to focus on two critical differentiation processes—metacyclogenesis and epimastigogenesis—emphasizing their environmental triggers, metabolic regulation, and roles in parasite transmission and life cycle progression.
The manuscript herein presented, might be interesting and important since there is an urgent need to evaluate the differentiation stages within the vector in order to better search for therapeutic and preventive measures against this disease. However, many issues have to be resolved.
- English language used as well as grammatic mistakes have to be carefully revised.
- In the abstract, extracellular vesicles are mentioned as a modulator of parasite differentiation and transitions but afterwards is not discussed.
- In line 373 a box is mentioned but it is confusing what the authors mean
- Table 2 is incomplete
- In line 375 the name references is written in Portuguese
- The description of the different items related to the differentiation are superficial and do not offer new insights or even ways of developing further in this discussion.
- So, in general, the manuscript has to be rewritten and discussed in more detail.
Author Response
Dear Editor and Reviewers,
We would like to sincerely thank you for the time and effort dedicated to reviewing our manuscript entitled “Extracellular Vesicles Promote Epimastigogenesis in Trypanosoma cruzi.” We greatly appreciate the constructive comments and suggestions provided, which have been invaluable in improving the clarity, structure, and scientific quality of our work.
Following the reviewers’ recommendations, we have thoroughly revised the manuscript. Major changes include a clearer description of the experimental design and statistical analyses; expanded discussion linking extracellular vesicle signalling to differentiation mechanisms; Improved English language and figure labelling throughout the text.
Below, we provide a detailed, point-by-point response to each comment, indicating how and where the manuscript has been modified. All changes have been highlighted in the revised version for ease of reference.
We hope that the revised manuscript now meets the journal’s standards and addresses all reviewers’ concerns. We remain at your disposal for any further clarification.
With our best regards,
Marcel I Ramirez, PhD on behalf of all authors
Instituto Carlos chagas
Curitiba
Brazil
Reviewer 2
Comments and Suggestions for Authors
The aim of this study has been to focus on two critical differentiation processes—metacyclogenesis and epimastigogenesis—emphasizing their environmental triggers, metabolic regulation, and roles in parasite transmission and life cycle progression.
The manuscript herein presented, might be interesting and important since there is an urgent need to evaluate the differentiation stages within the vector in order to better search for therapeutic and preventive measures against this disease. However, many issues have to be resolved.
English language used as well as grammatic mistakes have to be carefully revised.
In the abstract, extracellular vesicles are mentioned as a modulator of parasite differentiation and transitions but afterwards is not discussed.
In line 373 a box is mentioned but it is confusing what the authors mean
Table 2 is incomplete
We appreciate this remarks and have performed a through english language and style revision throughout the manuscript to enhance clarity and coherence .
In line 375 the name references is written in Portuguese
We have revised this section to clarify the purpose of the box and improved cross-referencing within the text.
The description of the different items related to the differentiation are superficial and do not offer new insights or even ways of developing further in this discussion.
We acknowledge this limitation. The paucity of experimental data on the molecular mechanisms underlying differentiation, particularly epimastigogenesis, restricts further elaboration. We have nonetheless strengthened this discussion and added a “Final Considerations” section highlighting open questions regarding T. cruzi development within triatomine vectors.
So, in general, the manuscript has to be rewritten and discussed in more detail.
Reviewer 3 Report
Comments and Suggestions for Authors
In this review article, Sana et al., describe Trypanosoma cruzi's developmental plasticity, enabling survival in diverse environments of insect vectors and mammalian hosts. They focus on two differentiation processes, metacyclogenesis and epimastigogenesis, triggered by environmental factors like nutrient starvation, pH, and temperature. These processes regulate parasite transmission and adaptation within the insect vector's gut. The review highlights metabolic and signaling differences and the role of extracellular vesicles in parasite communication and immune evasion. Understanding these mechanisms offers insights for disrupting the T. cruzi life cycle and controlling Chagas disease.
The manuscript is well written. However, I have some suggestions and hope to contribute to improve the quality of the paper.
Regarding the introduction section, I believe it would be improved by including more detailed recent findings on the genetic diversity and newly identified morphological forms of Trypanosoma cruzi. The authors might add insights into the molecular mechanisms governing life cycle transitions between replicative and infective stages. Incorporating advances in understanding parasite-host interactions at the cellular and immune levels would strengthen the context.
In addition, addressing current gaps in knowledge related to parasite adaptation in both insect and mammalian hosts could enhance the narrative.
In the topic #2 (biological differentiation and morphogenesis), the athors may improve section by clarifying the sequence of differentiation events, addressing existing controversies. This section would include more recent molecular insights into the mechanisms driving epimastigogenesis. Adding a clearer comparison of in vivo vs. in vitro findings would strengthen understanding.
Regarding the topic "Environmental Cues: Temperature, Stress, and Nutrient Sensing" the authors could improve this sectin by integrating more recent findings on how extracellular vesicles modulate differentiation processes, particularly epimastigogenesis. They should expand discussion on the complex interactions between parasite, microbiota, and insect vector physiology. Including insights into signaling pathways activated by environmental cues like temperature and nutrient availability would strengthen the analysis.
Regarding the topic "Autophagy and the Regulation of Metacyclogenesis" the authors could improve this section by incorporatng more recent molecular insights into the signaling pathways linking autophagy with metacyclogenesis regulation. It would be interesting to discuss the role of key proteins like TcVps34 and TcVps15 more comprehensively. Finally, integrating advances in understanding how autophagy influences protease activation and parasite infectivity would strengthen the analysis.
As I stated above, the manuscript is well written and cover an interesting topic in parasitology. I have made some suggestions and hope to contribute.
Author Response
Dear Editor and Reviewers,
We would like to sincerely thank you for the time and effort dedicated to reviewing our manuscript entitled “Extracellular Vesicles Promote Epimastigogenesis in Trypanosoma cruzi.” We greatly appreciate the constructive comments and suggestions provided, which have been invaluable in improving the clarity, structure, and scientific quality of our work.
Following the reviewers’ recommendations, we have thoroughly revised the manuscript. Major changes include a clearer description of the experimental design and statistical analyses; expanded discussion linking extracellular vesicle signalling to differentiation mechanisms; Improved English language and figure labelling throughout the text.
Below, we provide a detailed, point-by-point response to each comment, indicating how and where the manuscript has been modified. All changes have been highlighted in the revised version for ease of reference.
We hope that the revised manuscript now meets the journal’s standards and addresses all reviewers’ concerns. We remain at your disposal for any further clarification.
With our best regards,
Marcel I Ramirez, PhD on behalf of all authors
Instituto Carlos chagas
Curitiba
Brazil
Reviewer 3
Comments and Suggestions for Authors
In this review article, Sana et al., describe Trypanosoma cruzi's developmental plasticity, enabling survival in diverse environments of insect vectors and mammalian hosts. They focus on two differentiation processes, metacyclogenesis and epimastigogenesis, triggered by environmental factors like nutrient starvation, pH, and temperature. These processes regulate parasite transmission and adaptation within the insect vector's gut. The review highlights metabolic and signaling differences and the role of extracellular vesicles in parasite communication and immune evasion. Understanding these mechanisms offers insights for disrupting the T. cruzi life cycle and controlling Chagas disease.
The manuscript is well written. However, I have some suggestions and hope to contribute to improve the quality of the paper.
Regarding the introduction section, I believe it would be improved by including more detailed recent findings on the genetic diversity and newly identified morphological forms of Trypanosoma cruzi. The authors might add insights into the molecular mechanisms governing life cycle transitions between replicative and infective stages. Incorporating advances in understanding parasite-host interactions at the cellular and immune levels would strengthen the context.
In addition, addressing current gaps in knowledge related to parasite adaptation in both insect and mammalian hosts could enhance the narrative.
We agree entirely.A new section now discusses T cruzi genetic diversity and its potential impact on parasite development in the vector.The concluding section also highlights current knowledge gaps on adaptations in both hosts.
In the topic #2 (biological differentiation and both hosts) the athors may improve section by clarifying the sequence of differentiation events, addressing existing controversies. This section would include more recent molecular insights into the mechanisms driving epimastigogenesis. Adding a clearer comparison of in vivo vs. in vitro findings would strengthen understanding.
We acknowledge that molecular mechanisms controlling epimastigogenesis remain poorly understood. The specific signalling pathways and genes involved have yet to be fully identified. These questions form part of ongoing doctoral research in our group (A. Sana), which investigates the role of EVs in this process.
Regarding the topic "Environmental Cues: Temperature, Stress, and Nutrient Sensing" the authors could improve this sectin by integrating more recent findings on how extracellular vesicles modulate differentiation processes, particularly epimastigogenesis. They should expand discussion on the complex interactions between parasite, microbiota, and insect vector physiology. Including insights into signaling pathways activated by environmental cues like temperature and nutrient availability would strengthen the analysis.
We appreciate this insight. While our preliminary data suggest that EVs promote epimastigote formation, the underlying mechanisms remain to be elucidated. We therefore limited speculation but added clarifications on current evidence and open questions related to environmental signalling and EV involvement.
Regarding the topic "Autophagy and the Regulation of Metacyclogenesis" the authors could improve this section by incorporatng more recent molecular insights into the signaling pathways linking autophagy with metacyclogenesis regulation. It would be interesting to discuss the role of key proteins like TcVps34 and TcVps15 more comprehensively. Finally, integrating advances in understanding how autophagy influences protease activation and parasite infectivity would strengthen the analysis.
We have incorporated recent studies describing the functions of TcVps34 and TcVps15 and their relevance to autophagy and metacyclogenesis, thereby strengthening this section’s molecular focus.
As I stated above, the manuscript is well written and cover an interesting topic in parasitology. I have made some suggestions and hope to contribute.
Concluding Remarks
We have substantially revised the manuscript in response to reviewers’ comments, improving clarity, depth, and structural coherence. New sections now address T. cruzi’s genetic diversity and its potential influence on parasite differentiation within the insect vector. The discussion integrates updated information on vector physiology, microbiota interactions, and the potential modulatory role of extracellular vesicles. Finally, we added a comprehensive “Concluding Remarks” section that highlights persisting knowledge gaps and proposes future research directions aimed at elucidating the molecular mechanisms governing T. cruzi development and transmission.
Round 2
Reviewer 2 Report
Comments and Suggestions for Authors
The queries have been addresses therefore the manuscript can be accepted
Reviewer 3 Report
Comments and Suggestions for Authors
I thank the authors for their efforts to improve the quality of the manuscript